# Basal Cell Carcinoma—A Retrospective Descriptive Study Integrated in Current Literature

**DOI:** 10.3390/life13030832

**Published:** 2023-03-19

**Authors:** Carmen Giuglea, Andrei Marin, Iulia Gavrila, Alexandra Paunescu, Nicoleta Amalia Dobrete, Silviu Adrian Marinescu

**Affiliations:** 1Pastic Surgery Department, Faculty of Medicine, “Carol Davila” University of Medicine and Pharmacy, 020021 Bucharest, Romania; 2Plastic Surgery Department, “St. John” Hospital, 042122 Bucharest, Romania; 3Pathology Department, “St. John” Hospital, 042122 Bucharest, Romania; 4Hematology Department, Ploiesti County Hospital, 100576 Ploiesti, Romania; 5Plastic Surgery Department, “Bagdasar Arseni” Hospital, 041915 Bucharest, Romania

**Keywords:** skin cancer, basal cell carcinoma, skin surgery

## Abstract

Basal cell carcinoma (BCC) is considered to be the most common cancer in humans. It has a slow growth rhythm, and for this reason, metastases are rare. For our retrospective study, we selected 180 patients from those who underwent surgery for a variety of skin tumours between January 2019 and August 2022 and whose histopathological examination revealed basal cell carcinoma. All surgeries were performed by plastic surgeons at the “St. John” hospital in Bucharest. The aim of this article is to provide observational data regarding BCC—in terms of histopathology and diagnostic and therapeutic management and to integrate these data into the current knowledge of this pathology.

## 1. Introduction

Basal cell carcinoma is considered to be the most common malignancy in the white population [1]. This pathology has different incidences around the world—depending on race and geographic influences—with the highest incidence in Australia and the lowest in Africans from Kenya [2].

The pathogenesis of BCC involves the activation of the hedgehog intracellular signalling pathway, which is responsible for cell growth and division. There are several mutations which can occur in different genes: some with suppressive/inhibiting roles (PTCH1, SUFU, p53) and others which activate tumour formation (SMOm) [3].

However, this type of cancer has a slow growth rate, usually resulting in a local invasion of surrounding tissue, with very few metastases [4].

BCC has a predominance in male patients according to Asgari et al. [5]. The major risk factor is considered to be sun exposure (UVA and UVB radiation), with a predilection for exposed skin areas (head and neck) and a fair skin complexion (Fitzpatrick types 1 and 2) [3,5]. Immunosuppression is also a significant risk factor, as patients with organ transplants are 5 to 10 times more likely to develop BCC [3]. Other risk factors that deserve mention are a family history of skin cancer, artificial tanning, photosensitising drugs, childhood sunburn, ionising radiation and chemicals [6,7,8,9,10].

According to WHO Classification of skin tumours, BCC includes the following histopathological subtypes: superficial BCC, nodular (solid) BCC, micronodular BCC, infiltrating BCC, sclerosing/morphoeic BCC, fibroepithelial BCC, BCC with adnexal differentiation, basosquamous carcinoma, BCC with sarcomatoid differentiation and pigmented BCC [11].

There are several diagnostic tools, which include dermoscopy, as well as clinical microscopy [12] and confocal microscopy [13]; nonetheless, the final diagnosis is provided from the histopathology examination. Other diagnostic tools include high-resolution ultrasonography, optical coherence tomography, Raman spectroscopy, terahertz pulse imaging and reflectance confocal microscopy [14,15].

## 2. Materials and Methods

The inclusion criteria for the patients in our study were represented by a positive diagnosis for BCC, taken from histopathology reports of patients that were operated on in St. John’s Hospital between January 2019 and August 2022. All patients that were operated on in this time frame for skin lesions with a positive histopathologic result for BCC were selected for this study (including patients that were operated on for multiple lesions in which at least one turned out to be a BCC). Exclusion criteria were represented by operated skin lesions for which there was no histopathologic report and singular skin lesions which had other results on the histopathologic report (SCC, melanoma).

The variables extracted from these reports were represented by age and sex of the patient, experience of the operator, tumour localisation, date of the operation, size of the excised specimen (length, width and depth), largest diameter of the BCC, markings of the specimen for orientation, histopathological type (infiltrative/nodular/micronodular/superficial/basosquamouspigmented), if the tumour was ulcerated, if the margins were clear of tumour, if more tumours were excised during the same operation (and if those were benign or malignant). 

From the medical file of the patients with BCC, we extracted the following data: type of surgery (direct closure/skin graft/flap), if the patient came for a relapse or for a BCC at the first excision, patients which were re-excised for incomplete first excision and personal medical history for each patient.

The data were processed by using IBM Statistical Package for Social Science 25 (SPSS).

## 3. Results

Our study included a total of 180 patients with a total of 211 basal cell carcinomas operated (155 patients had 1 BCC, 22 patients had 2 BCC, 2 patients had 3 BCC and 1 patient had 5 BCC). The sex distribution in our group was 88 women (48.9%) and 92 men (51.1%). The median age of the women’s group was 73 years, while for men the median was 73.5 years. Age/sex distribution is represented in the following histogram (Figure 1).

Between January 2019 and August 2022, there were 180 patients operated for BCC in our hospital. Figure 2 shows the percentage of patients operated each year.

Table 1 reflects the distribution of BCC related to age. Only 5.6% of the total number of patients with BCC was under the age of 50. 

The 211 BCC lesions were classified based on localisation according to the following table (Table 2).

Out of the 211 lesions with BCC, 206 of them had recordings of the surface area. The median surface area of the lesions was 2.81 cm^2^, with the following difference between the 2 genders: 2.341 cm^2^ (IQR: 3.32 cm^2^) for women and 3.42 cm^2^ (IQR: 4.09 cm^2^) for men.

Although our study revealed a difference between men and women in terms of the median surface area of the lesions (3.42 cm^2^ for men and 2.341 cm^2^ for women), this result lacked statistical power (*p* = 0.094). We also analysed whether the surgeon opted to excise the tumour in one single piece or if he/she excised the tumour and the margins separately (in order to evaluate if there are residual tumour cells). In 27 cases out of 211 (12.8%), the margins were separately excised, while for the 184 remaining cases, the tumours were excised entirely without separate margins.

Another aspect which we took into consideration was the marking of the tumour for histopathology orientation. Of the lesions, 144 (68.2%) were not marked, 15 (7.1%) were marked in one extremity and 52 (24.6%) had 2 different extremities marked for proper orientation.

The BCC subtypes were also noted—the lesions were described as infiltrative, nodular, micronodular, superficial, pigmented and basosquamous. The other subtypes were not found in our histopathological results. In 11 cases, there was no mention of a histopathological subtype. In most cases, these histopathological subtypes were mixed (2 or more subtypes were present in a single BCC lesion). In Table 3, we presented the total number of each subtype found (either alone or mixed with other subtypes), and in Table 4, we analysed the mixed subtypes separate from the lesions which had only a single histopathological subtype. 

Ulceration of the BCC in evolution was also analysed; 132 of 211 (62.6%) lesions presented ulceration. The results are presented in Table 3 and Table 4.

One more aspect which was taken into consideration was whether the same patient that was operated for a BCC had other skin tumours which were operated at the same time of the BCC excision. In 139 cases (65.9%), there were no other tumours other than a single BCC. In 49 cases (23.2%), there were other benign tumours excised, while in 23 cases (10.9%), there were other malignant tumours excised simultaneously with the BCC. The separate benign tumours which were excised together with the BCC lesions were one of the following: haemangiomas, sebaceous nevi, sebaceous keratosis, papillomas or simple tissue fragments that presented histological modifications. The other malignant/premalignant tumours included squamous cell carcinoma, actinic keratosis or at least one other BCC lesion on the same patient.

The *T* value in the TNM system was also analysed, with the results presented in Table 5.

We also divided the 211 excisions based on the type of surgery performed, and the results are depicted in Table 6.

In terms of tumour margins, we noticed that 36 patients (17.1%) had positive margins. (Table 7).

Of the operated patients, 17 presented themselves for a recurrence for BCC, while 194 had a primary BCC (Table 8).

We analysed the group of patients with BCC recurrence to see whether a consultant with more than 10 years of experience had less cases of BCC recurrence compared to a specialist with less than 10 years of experience. Out of 87 patients, 4 patients (4.6%) had a BCC recurrence in the specialist group, while in the consultant group, 13 (10.4%) patients out of 124 had a recurrence. There was no statistical difference between the two types of doctors (X^2^ = 2.39, *p* = 0.112) (Table 9 and Table 10).

We analysed whether the surface area of the excised tumours correlated with the marking decision of the specimen. Table 10 reflects this decision; tumour marking was performed for BCC with a higher surface area. The marking of the tumour was statistically significantly associated with the median surface area (X^2^ = 2.8, *p* < 0.001).

We compared the difference between the specialist and consultant in terms of the median surface area of the operated BCC. A total of 206 BCC lesions had recordings of the surface, with a median of 2.81 cm^2^. The specialists operated a total of 87 BCC lesions; 2 of them had no recordings of surface area, while the remaining 85 lesions had a median surface area of 2.55 cm^2^ (IQR = 3.58). The consultants operated a total of 124 lesions; 3 of them had no recordings of the surface area, and the remaining 121 BCCs had a median surface area of 3.2 cm^2^ (IQR = 3.53). There was no statistical difference between the surface areas of the lesions operated by the specialists and those operated by the consultants (X^2^ = 2.81, *p* = 0.203).

We also analysed whether the depth of the tumour could be associated with the recurrence incidence. Table 11 reflects these results; the tumour depth could not be associated with the cases in which recurrence occurred (X^2^ = 5, *p* = 0.665).

We also evaluated if the surface area of the lesions can be associated with the positive/negative margins. The median surface area for negative margins was 2.85 cm^2^ (IQR 3.78 cm^2^) and the mean: 4.73 cm^2^, SD 6.5 cm^2^. The median surface area for positive margins was 2.54 cm^2^ (IQR 3.29 cm^2^)and the mean 4.39 cm^2^, SD 5.5 cm^2^. Therefore, the surface area of the lesion could not be associated with either type of margins (X^2^= 2.81, *p* = 0.463).

The cases with recurrence were analysed to see whether they had positive margins. A higher percentage of relapse was observed in the cases in which the margins were positive (*p* = 0.003). These results are presented in Table 12.

Finally, all comorbidities of the patients were noted and analysed in Table 13.

## 4. Discussion

Although the female/male ratio inclined the balance in favour of men in our study, there was only a small difference between sexes (the male:female ratio in our study was 1.045:1). The age distribution indicated that BCC appeared frequently in the 7th decade (with over 90% of all BCC that were excised being in the case of patients above 50 years old), with the median age being slightly higher in the male patients. 

The number of patients was significantly smaller in 2020 compared to 2019 due to the COVID-19 pandemic, but this number was still higher compared that of 2021 (when the hospital was declared a COVID hospital, and no other pathologies were allowed for treatment). The year 2022 showed a mild comeback (taken into consideration that the first few months were still under strict COVID regulation and that the results of the study were only until August 2022). This is due to the fact that during the COVID pandemic, many patients with chronic pathologies have delayed seeking medical care [14].

Almost 1/3 of the operated BCC were located on the nose, with more than 75% of all BCC being located at the level of the head. This clearly suggests a higher frequency of this type of malignancy at the more sun exposed regions and is consistent with the research presented by Costache et al. and Asilian et al. [16,17].

BCC is a tumour that can vary in size: from small lesions to giant ones, with giant BCC being over 5 cm in diameter [18]. Although we did not have a significant statistical difference between men and women in term of the size of the lesions, the men presented with higher surface-area BCC. This could be explained by the fact that men generally seek professional healthcare later than women (allowing such tumours to grow larger). Based on the size of the tumour according to the TNM scale, almost 80% of the patients were included in T1, and more than 95% are below T2, which indicates that most of these tumours are excised before an important local invasion.

Tumour markings and tumour margins are extremely important in the management of BCC; our study showed that while separate margins were taken in only 12.8% of the cases, almost 1/3 of all excised tumours were marked for orientation. Tullet et al. questioned whether the marking sutures for orientation were useful for further management of BCC, and in his study, only one case was re-operated based on those margins (0.2%). They concluded that routine marking is not necessary and should be performed in case of ill-defined lesions or for histopathological types with high risk [19].

In our study, a positive correlation between surface area and tumour marking was shown; this indicates that tumour marking was performed for BCC with higher surface area due to the operator’s suspicion that the bigger tumours might not be completely excised, and a new surgical intervention might be needed.

According to Costache et al., there are five subtypes of BCC based on histopathology: nodular, infiltrative/morpheaform, superficial, pigmented and fibroepithelioma of Pinkus [16]. Our study is consistent with the frequencies of these histopathology subtypes in this article, with the nodular type reaching 75% of all cases and the infiltrative type reaching more than 80% of all cases. In our study, all histopathological subtypes were present, both alone and mixed, except for the pigmented subtype, which appeared only in mixed subtype lesions. The majority of the CBC lesions (69.67%) had mixed histopathological subtypes.

Another study conducted by Fung-Soon Lim et al. divided the histopathological subtypes of BCC based on the risk of subclinical extension into two categories: high risk (morpheaform, infiltrative, metatypical, mixed and superficial) and low risk (basosquamous, micronodular, nodular and unspecified) [20]. These aspects are important for a clinician to evaluate the relapse risk after tumour excision. 

There are two aspects which can be researched and improved when considering BCC: one is represented by the diagnostic tools, and the other is represented by the therapeutic options. The limitation of our study is represented by the diagnostic and therapeutic options. We used clinical diagnosis and classical surgical excision for all lesions. There are, however, more modern diagnostic tools and therapeutic options which can achieve good results.

With the development of modern medical engineering, the correct diagnosis and the appropriate size of the BCC (in terms of depth and surface area) can now be evaluated even before the surgical excision. This has a major advantage due to the fact that it can reduce significantly the relapse rate in case of BCC (thus improving the quality of life of patients, who will undergo only one operation instead of two or more). Niculet et al. consider that the combination of optical coherence tomography (OCT) and reflectance confocal microscopy (RCM) can provide useful information for both depth and horizontal extension of a tumour and could be used prior to surgery in order to explore subclinical extension [21]. This is especially useful in the case of high risk histopathological subtypes of BCC, where due to an incomplete excision, relapse is a major concern.

RCM offers information about blood vessels at the level of the BCC in terms of density, size, and flow intensity. BCC, compared to benign tumours or normal skin, presents a peripheral stroma which has a higher density of microvessels [22]. 

From a histopathological perspective, the differential diagnosis of basal cell carcinoma includes both benign and malignant lesions. Trichoepithelioma and trichoblastoma represent the main benign tumours with a basaloid morphology that need to be taken into consideration. These tumours, however, only rarely present immunoexpression for BerEp4 and CD10, with the latter being more frequently positive in the peritumoural stroma of trichoepithelioma [23]. A microcystic or pseudoglandular morphology of BCC may also pose differential diagnosis problems with microcystic schwannoma, from which it can be easily differentiated by using an immunohistochemical panel containing S100, SOX10 and BerEP4 [24]. From the malignant category, one should always have in mind the option of a cutaneous metastasis. A basosquamous carcinoma must be differentiated from a basaloid squamous cell carcinoma (SCC), a keratoacanthoma or an adenoid cystic carcinoma. Both SCC and BCC can express p63; however, SCC usually expresses EMA, while BCC characteristically expresses BerEP4 [25,26]. 

In terms of therapeutic options, the surgical excision with negative margins represents the treatment of choice for plastic surgeons. In our study, the majority of the operated tumours were closed directly, which supports the idea that BCC is a slow-growing tumour and the surgical management can be performed in most cases under local anaesthesia without complications. While it is generally known that a skin graft may be easier to perform, the aesthetic result after a skin graft is inferior to the local flap. With 16% of all operations performed being a local flap (compared to 10% for skin graft), one can observe the tendency of the plastic surgeons to achieve coverage by using more aesthetic methods. This option could be influenced by the fact that the majority of the operations were performed at the level of the head, where good aesthetic results are expected.

An important surgical option is Mohs micrographic surgery (MMS), which is considered to be the standard of care for BCC and skin cancers in general. Although this technique is conservative and preserves as much as possible from the surrounding tissue, due to its high costs, it has some indications when MMS is considered appropriate: cases with risk of disfigurement; large malignant tumours; tumours with aggressive histopathological subtype or with poorly defined margins; recurrent tumours and skin cancers caused by genetic predisposition [27].

There are also some alternative options to surgery which can be used in the case of BCCs. However, it is mandatory that a correct selection of patients is performed when choosing these alternative options in order not to risk a recurrence. Among the non-surgical treatments which can be used, cryosurgery, ablative CO_2_ laser, 5-fluorouracil and imiquimod are the most frequently used therapies, which are performed specifically in dermatology practices. The advantages of these options are represented by the following arguments: they are non-invasive/minimally invasive; they achieve a better aesthetic result; and they can be used at the same time in case of multiple tumours. The disadvantages consist in the fact that they may be useful only for small tumours; they may have a higher risk of recurrence; and in some cases, more than one treatment session might be needed (compared to surgical excision, when all is performed in a single, more invasive stage).

Scurtu et al. reported superior cosmetic results from using these types of treatments compared to surgery, with a recurrence rate of under 1% [28]. Thompson et al. sustain the idea that although surgical excision has lower recurrence rates, non-surgical treatments have superior aesthetic results with acceptable recurrence rates [29]. For unresectable BCC or metastatic BCC, vismodegib—a kinase inhibitor—remains the primary option for treatment [30].

Among our operated patients, ~34% of them presented other skin tumours which were operated at the same time of the surgical excision of the BCC (23% benign and 11% malignant). This percentage of patients who had other skin tumours could be significantly higher (because not all patients want all their tumours operated in one surgery).

There was no association between the experience of the surgeon and the recurrence rate. Paradoxically, the consultants (doctors with over 10 years of experience) seemed to have more cases of recurrent BCC compared with specialists (doctors with less than 10 years of experience). This could be due to the fact that the consultants had more cases (124 cases vs. 87 cases), and these cases were more difficult compared to the cases solved by specialists (the median surface area of the BCC lesions was 2.55 cm^2^ in the case of specialists compared to 3.2 cm^2^ in the case of consultants).

In our study, the positive/negative margins of the specimen could not be associated with the surface area of the tumour, and neither could the recurrence rate with the tumour depth. This proved that even though large tumours (in size or depth) could theoretically be more susceptible to incomplete excision followed by recurrence (as most tumours are located at the level of the face, where the surgeon tries to be as conservative as possible), this was not the case in our study. However, an incomplete excision with positive margins has been statistically proven to cause a recurrent BCC more frequently, which would need a new surgical intervention. For this reason, either MMS or an extemporaneous histopathological examination could reduce the risk of an incomplete excision with positive margins.

With regard to comorbidities, 66.8% of all patients presented some type of cardiovascular pathology. This high percentage is, however, explicable due to the median age of the patients in our study, as the seventh decade has a high probability of having such comorbidities. The hematologic, pulmonary and infectious comorbidities were not so frequently seen in the personal history of the patients with BCC; renal comorbidities, diabetes and digestive comorbidities were, however, found in over 15% of all patients. Renal transplant is frequently associated in the literature with BCC, which could explain the high incidence of BCC in patients with renal comorbidities [31,32].

The future management of BCC is probably based on gene analysis and biomarkers that influence the prognosis of BCC, with p16 being one of the genes involved in the pathogenesis of human BCC [33,34].

## 5. Conclusions

BCC is a pathology which should be approached in a multidisciplinary team. Our study reveals aspects related to margins, surface areas and surgical treatment approach used in our clinic.

## Figures and Tables

**Figure 1 life-13-00832-f001:**
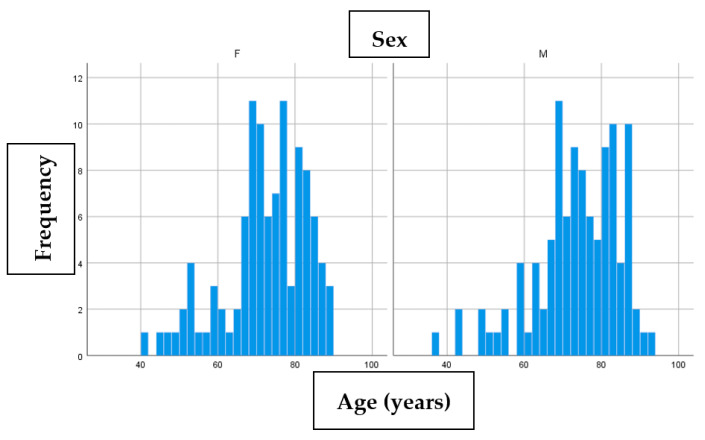
Age/sex distribution for BCC.

**Figure 2 life-13-00832-f002:**
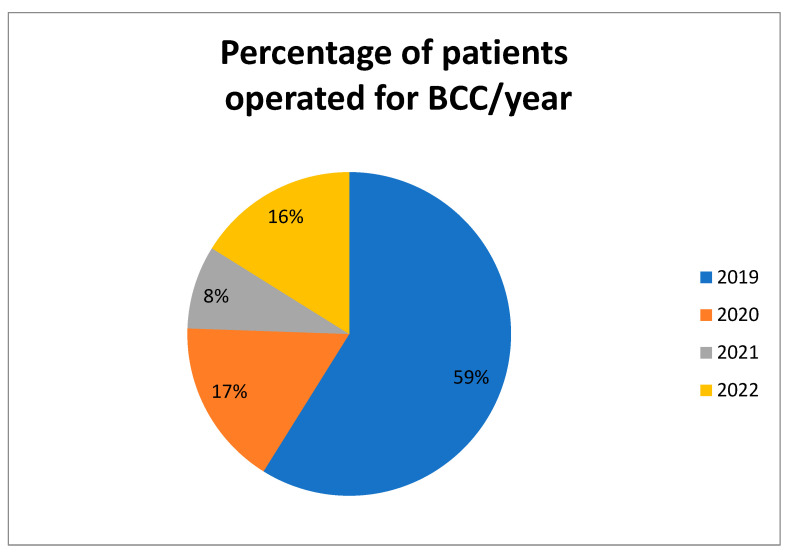
The percentage of patients operated each year.

**Table 1 life-13-00832-t001:** Distribution of BCC related to age.

Percentage of Patients </> 50 Years Old		
Patient’s Age	No. of Patients	%
≤50 years	10	5.6
>50 years	170	94.4
Total	180	100.0

**Table 2 life-13-00832-t002:** Localisation of BCC.

Localisation	Number	Percentage
Nose	65	30.8
Scalp	14	6.6
Fronto-temporal area	25	11.8
Genio-maseterian area	27	12.8
Periocular region	20	9.5
Perioral/mental region	7	3.3
Auricular	13	6.2
Neck	18	8.5
Trunk	14	6.6
Upper limbs	6	2.8
Lower limbs	2	0.9
Total	211	100.0

**Table 3 life-13-00832-t003:** BCC subtypes (absolute values).

Histopathological Subtype	Present	Absent/ Not Mentioned	Total
Infiltrative	170 (80.6%)	41 (19.4%)	211 (100%)
Nodular	157 (74.4%)	54 (25.6%)	
Micronodular	8 (3.8%)	203 (6.2%)	
Superficial	10 (4.7%)	201 (95.3%)	
Basosquamous	6 (2.8%)	205 (97.2%)	
Pigmented	3 (1.4%)	208 (98.6%)	

**Table 4 life-13-00832-t004:** BCC subtypes (isolated and combined).

Histopathological Subtype	Number and Percentage
Infiltrative only	28 (13.27%)
Nodular only	17 (8.06%)
Micronodular only	2 (0.95%)
Superficial only	4 (1.9%)
Basosquamous only	2 (0.95%)
Mixt (2 or more of the above)	147 (69.67%)
Missing	11 (5.21%)
Total	211 (100%)

**Table 5 life-13-00832-t005:** *T* values for BCC lesions.

*T* Value	Number of Lesions	Percentage
0	2	0.9
1	131	62.1
2	25	11.8
3	8	3.8
Total present	166	78.7
Missing	45	21.3
Total	211	100.0

**Table 6 life-13-00832-t006:** Type of surgery.

Operation Type	Number	Percentage
Direct suture	149	70.6
Skin graft	22	10.4
Local flap	35	16.6
Regional flap	5	2.4
Total	211	100.0

**Table 7 life-13-00832-t007:** Tumour margins.

Margin Type	Number	Percentage
Negative margins	175	82.9%
Positive margins	36	17.1%
Total	211	100%

**Table 8 life-13-00832-t008:** Type of tumour.

Recurrence		
Type of Tumour	Number	Percentage
Primary tumour	194	91.9%
Recurrent tumour	17	8.1%
Total	211	100%

**Table 9 life-13-00832-t009:** Recurrence related to operator experience.

Recurrence Rate in Relation to Operator’s Experience				
			Specialist	Consultant
Recurrence	Without recurrence	Count	83	111
		Percentage	95.4%	89.6%
	With recurrence	Count	4	13
		Percentage	4.6%	10.4%
Total		Count	87	124
		Percentage	100%	100%

**Table 10 life-13-00832-t010:** Association between tumour marking and surface area.

Case Processing Summary						
	Valid		Missing Cases		Surface Area (cm^2^)	
Tumour marking	N	Percentage	N	Percentage	Median	IQR
Unmarked	142	98.6%	2	1.4%	2.23	2.97
1 marked pole	13	86.7%	2	13.3%	3.5	3.11
2 marked poles	51	98.1%	1	1.9%	4.68	5.35

Each row tests the null hypothesis that the Sample 1 and Sample 2 distributions are the same. Asymptotic significances (2-sided tests) are displayed. The significance level is 0.05. Significance values have been adjusted by the Bonferroni correction for multiple tests.

**Table 11 life-13-00832-t011:** Association between recurrence and tumour depth.

Case Processing Summary						
Recurrence	Valid		Missing		Depth	
	N	Percentage	N	Percentage	Depth Median	IQR
Cases without recurrence	164	84.5%	30	15.5%	5 mm	4 mm
Cases with recurrence	15	88.2%	2	11.8%	5 mm	5 mm

**Table 12 life-13-00832-t012:** Association between tumours with invaded margins and relapses.

			Relapsed Lesions		Total
			No Relapse	Relapsed	
Margins	No tumoral invasion	Count	166	9	175
		% within Margins	94.9%	5.1%	100.0%
	Tumoral invasion	Count	28	8	36
		% within Margins	77.8%	22.2%	100.0%
Total		Count	194	17	211
		% within Margins	91.9%	8.1%	100.0%

**Table 13 life-13-00832-t013:** Personal history of patients with BCC.

Complications	Number (Percentage)		
	Without	Present	Total
Cardiovascular	70 (33.2%)	141 (66.8%)	211 (100%)
Diabetes	168 (79.6%)	43 (20.4%)	
Infectious	204 (96.7%)	7 (3.3%)	
Pulmonary	198 (93.8%)	13 (6.2%)	
Digestive	172 (81.5%)	39 (18.5%	
Hematologic	205 (97.2%)	6 (2.8%)	
Renal	177 (83.9%)	34 (16.1%)	

## Data Availability

The datasets used and/or analyzed during the present study are available from the corresponding author upon reasonable request.

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
