# Peer review of "Basal Cell Carcinoma—A Retrospective Descriptive Study Integrated in Current Literature"

_life, 2023, doi:10.3390/life13030832_

Round 1
Reviewer 1 Report
Title - Basal cell carcinoma – a retrospective descriptive study
General Comments
The authors present a retrospective descriptive study of BCC at a Hospital. The authors did not refer the aim of the manuscript (MS). The MS has poor organization and presentation of data; the authors did not use the data to present the most clinically relevant information. In this present form the MS is not suitable for publication.
Specific Comments
Introduction
1. Line 27 – Reference (1) is an old reference (2005), considering the subject. The authors should have used more recent ones.
2. Line 36 – “(female/male ratio = 2/1)” - Who says this? Most papers do not report such a rate.
Materials and Methods
3. Line 51 – “January 2019 and August 2022” – Were all the patients operated during this period, and meeting the inclusion and exclusion criteria, included in the study?
4. Line 55 - “infiltrative/nodular/superficial/ulcerated” – this classification was based on?
Results
5. Line 64 – According to the data presented, the total is 210 and not 211.
6. Line 72 – Table 1 - What is the point of this Table? Why include cumulative percent? Why copy the table directly from SPSS and put 3 equal columns?
7. Line 87 – Table 4 – Why use the SPSS statistics table? Comments 6 and 7 hold true for all the Tables of the MS.
8 – Line 98 – This data presentation does not make sense. The authors should have included one single Table including the various histological types.
9 – Line 109 – Table 12 – The authors could just present the figures, without any tables, and indicate which types of skin tumor were present.
10 – Line 141 – Table 22 - The authors are almost presenting raw data for the reviewer to analyze them. At this moment, the reader is lost.
Discussion
11 – Lines 165-166 – How do the authors relate the data presented in this sentence with the data obtained in their present study?
12 – Line 181 – “can grow to important sizes” – What does this mean?
13 – Lines 189-191 – What about the recurring cases? Had they smaller tumour margins?
14 – Lines 200-202 – This was not what the authors have describe in the Introduction section, nor is it the classification they used in their analysis.
15 – Lines 204-208 - In the present study, why didn't the authors use this classification?
16 – Lines 211-236 - I do not see the pertinence of discussing these subjects that are not directly related to the data presented in the manuscript.
17 – Line 254 – Again, it seems the authors are doing a review article on the diagnosis and therapy of BCC and not discussing the results of their own study.
18 – Line 275 - If there is no association, why do the authors next give explanations for this difference that was not significant?
19 – Lines 296-298 – What is the purpose of this sentence at the end of the discussion?
Author Response
The aim of the manuscript is to provide observational data regarding BCC - in terms of histopathology, diagnostic and therapeutic management and to integrate this data in the current knowledge of this pathology (the aim was added in the text). We have restructured the manuscript in order to make the article easier to follow.
Regarding the specific comments:
- the older reference was replaced by a newer one
- the female:male ratio was taken from this article (Basset-Seguin N, Herms F. Update in the Management of Basal Cell Carcinoma. Acta Derm Venereol. 2020;100(11):adv00140. doi:10.2340/00015555-3495) - I added this once again as reference.
- all the patients were operated during January 2019 and August 2022 - this was the inclusion criteria (exclusion criteria were patients with BCC operated outside this time frame)
- we extracted from the histopathological reports following subtypes, which are also present in literature (infiltrative/nodular/micronodular/superficial); the ulceration is not a subtype and therefore it was accounted separately as a tumour characteristic.
- There were 211 tumours. We have corrected all tables and data if there were any oversights in the original paper, it has now been corrected
- Modified as requested
- Modified as requested
- Modified all data in one table
- Modified as requested
- The data in the table was translated in written text to be easier to follow.
- In our study we had 92 male patients and 88 female patients with a male:female ratio of 1:1.045. I added this ratio in the text to enhance the fact that there a small difference between the sexes.
- We removed "can grow to important sizes" and left only the comment that if it is larger than 5cm in diameter, it is called giant BCC, according to Oudit et al.
- We added a correlation between tumours margins and recurrence in the results and discussed it in the discussion section
- The histopathologic subtypes used in our study were taken from each patient's file report. The common ground is represented by the main subtypes which are present in all classifications and in our study as well: nodular, infiltrative, superficial, micronodular. As we did not have any case from the subtypes pigmented and fibroepithelioma of Pinkus, we decided to only mention it in the discussion section.
- There are many classifications for histopathology subtypes of BCC. We only presented the subtypes which we had available in our reports.
- Due to the fact that our diagnostic tool was the clinical examination and the histopathology exam, we considered useful for the reader of the article to also have a brief description of the clinician's other options.
- We discussed our own findings but we also wanted to offer more insight on the diagnostic and therapeutic options of BCC. If needed, we can add to the title "BCC - a retrospective descriptive study integrated in current literature"
- The fact that there is no statistical association between experience and recurrence rate doesn't necessarily say that there is nothing to be discussed on the subject. We only observed that the recurrence rate was higher in the consultant group compared to the specialist group (which is exactly opposite to what one might expect), even though this didn't have statistical power. If larger number of operations would have been evaluated, perhaps this would have changed as well.
- The purpose of this sentence is to give a perspective on the future management of BCC in order to offer the reader not only the current diagnostic and therapeutic options, but also to show a glimpse of what the future has in store.
Reviewer 2 Report
A well written paper. No stand out new or unusual findings.
Page 37 Hands are not common sites for BCC presentation. Please correct. How many of the cases involved hands in this study?
Page 55 Ulceration when present is very common on all BCC subtypes. Ulceration is not a specific histological subtype of BCC.
Table 3 Neck sites should be separate from trunk. Note correct spelling for trunk.
Pages 225 and 229. Correct the spelling for trichoepithelioma.
Author Response
Thank you for your comments.
We have removed the word hands from the original text (only 2 patients/180 had BCC on their hands).
We have removed from the article the idea of ulceration to be a histological subtype and described it as a specific trait of different BCC in their evolution.
We have separated the neck and trunk into 2 categories and corrected the word trunk.
Corrections of the word trichoepithelioma were made.
Reviewer 3 Report
Thank you for allowing me to review this review article entitled "Recent advances in acral melanoma treatments and diagnosis".The work is well-conducted, and the authors provide detailed information about their management of cutaneous basal cell carcinoma in everyday clinical practice. Although in the article, too many tables make the results incomplete and not so presented. The work is valuable, but it needs to add something new to the Literature.
Author Response
Thank you for your review!
We have restructured the content, reducing the number of tables and making the information easier to follow. We highlighted the main findings and while the article might not have a high degree of novelty, it contains detailed and specific information about the subject.
Round 2
Reviewer 1 Report
Title - Basal cell carcinoma – a retrospective descriptive study
General Comments
The authors made changes in the manuscript but most major issues were not addressed.
Specific Comments
Introduction
1. Line 39 – “(female/male ratio = 2/1)” – The authors included a reference to justify sating this quite uncommon ratio. This reference is a review paper which included this statement as a citation of another article: “Trends in Basal Cell Carcinoma Incidence and Identification of High-Risk Subgroups, 1998-2012”. I went to look for this article and this was not sated in this article, a very different ratio is presented, since 54.8% of BCC patients studied were male.
Materials and Methods
2. Previous comment – “January 2019 and August 2022” – Were all the patients operated during this period, and meeting the inclusion and exclusion criteria, included in the study? For me it is still not clear if all the patients operated during this period and having a BCC were included in the study. There were no exclusion criteria?
3. Previous comment - “infiltrative/nodular/superficial/ulcerated” – this classification was based on? The authors have changed the classification, but still do not say (cite) which classification they based themselves on. How did they classify mixed cases (where more than one subtype is present)?
Results
4. Table 3 – The results concerning the histological subtypes continue to be presented in a confusing way. The authors should have subdivided the 211 BCCs in the histological subtypes, and the total should be 100%.
5. Lines 146-148 – Which were the other tumours mentioned?
6. Line 248 – The authors kept using the SPSS raw tables.
Discussion
7. Several important questions that I have raised in the previous version of the manuscript were not addressed.
Author Response
Thank you for the comments.
- I have replaced the epidemiologic findings with the information from the article you mentioned
- All patients during this time interval operated for skin lesion with a positive histopathologic report were included in this study. This also included patients operated for multiple lesions in which at least one turned out to be a BCC. Exclusion criteria was represented by operated skin lesions where there was no histopathologic report and singular skin lesions which had other results on the histopathological report (SCC, melanoma).
- We added the WHO Classification in the introduction
"According to WHO Classification of skin tumours, BCC includes following histopathological subtypes: superficial BCC, nodular (solid) BCC, micronodular BCC, infiltrating BCC, sclerosing/morphoeic BCC, fibroepithelial BCC, BCC with adnexal differentiation, basosquamous carcinoma, BCC with sarcomatoid differentiation and pigmented BCC [11].."
We only have 6 subtypes in our histopathological results (we added the basosquamous and the pigmented which were previously not mentioned, but they were described in the results in our reports). The other subtypes are not common findings and as we did not find them in our reports, we did not include them in our study. - The histopathological reports had mostly BCC with 2 or more subtypes (mixed). We made 2 separate tables - the existent table (where we added the pigmented and basosquamous subtypes) where we just counted every time a subtype was mentioned in a histopathology report and added a second table (where we presented where we had lesions with only one histopathological subtype and the rest were mixed subtypes in the same lesion). In the second table, we also added the reports where there were no subtypes mentioned and by adding all these numbers we achieved 100% (211 lesions).
- We added in the text "The separate benign tumours which were excised together with the BCC lesions were one of the following: hemangiomas, sebaceous nevi, sebaceous keratosis, papillomas, or simple tissue fragments that presented histological modifications. The other malignant/premalignant tumours included: squamous cell carcinoma, actinic keratosis or at least one other BCC lesion on the same patient."
- We removed all SPSS tables.
- The following questions were addressed in the previous discussion section, which received these answers in the last response. We will reply again to each.
11. How do the authors relate the data presented in this sentence with the data obtained in their present study? "In our study we had 92 male patients and 88 female patients with a male:female ratio of 1:1.045. I added this ratio in the text to enhance the fact that there a small difference between the sexes."
We presented the new epidemiologic data at point 1
12. “can grow to important sizes” – What does this mean?"
We removed "can grow to important sizes" and left only the comment that if it is larger than 5cm in diameter, it is called giant BCC, according to Oudit et al.
This question was previously answered and the statement in the article was rewritten.
13. What about the recurring cases? Had they smaller tumour margins?
We added a correlation between tumours margins and recurrence in the results and discussed it in the discussion section
Following phrase was added "The cases with recurrence were analyzed to see whether they had positive margins. A higher percentage of relapse was observed in the cases where the margins where positive (p=0.003). These results are presented in table 12."
14. This was not what the authors have describe in the Introduction section, nor is it the classification they used in their analysis.
We added in the introduction the WHO Skin tumours classification and added to our study 2 histopathological subtypes which we previously not mentioned - the pigmented and the basosquamous.
15. In the present study, why didn't the authors use this classification?
The WHO classification used now is novel (2018) and most comprehensive. Although not all histopathological subtypes were found in our study, we quantified in the first table all mentioning of each subtype and in the second table only mentioning of the individual subtype (plus the lesions which had mixed subtypes in the same lesion). The second table - is basically a quantification of each lesion (which either had a single histopathological subtype in the lesion, or mixed or the subtype was not mentioned).
17. I do not see the pertinence of discussing these subjects that are not directly related to the data presented in the manuscript. Due to the fact that our diagnostic tool was the clinical examination and the histopathology exam, we considered useful for the reader of the article to also have a brief description of the clinician's other options.
Thank you for your review. We change the title to "Basal cell carcinoma – a retrospective descriptive study integrated in current literature" in order to justify discussing such subjects.
We also added following statements to the article which justify discussing the subjects:
"The limitation of our study is represented by the diagnostic and therapeutic options. We used clinical diagnosis and classical surgical excision for all lesions. There are however more modern diagnostic tools and therapeutic options which can achieve good results."
18. Again, it seems the authors are doing a review article on the diagnosis and therapy of BCC and not discussing the results of their own study. We discussed our own findings but we also wanted to offer more insight on the diagnostic and therapeutic options of BCC.
We acknowledge that beside the personal data, we also present some observations related to data from the literature, with suitable references.
The editor also suggested a minimum of 30 references of recent articles and by removing the text, this demand could no longer be met. Please take these arguments into consideration.
18. If there is no association, why do the authors next give explanations for this difference that was not significant? The fact that there is no statistical association between experience and recurrence rate doesn't necessarily say that there is nothing to be discussed on the subject. We only observed that the recurrence rate was higher in the consultant group compared to the specialist group (which is exactly opposite to what one might expect), even though this didn't have statistical power. If larger number of operations would have been evaluated, perhaps this would have changed as well.
All results, whether having statistical power or not, should be commented in our opinion. In some cases, there are some causes why the statistical power was not achieved and these aspects should not be overseen.
19. What is the purpose of this sentence at the end of the discussion? The purpose of this sentence is to give a perspective on the future management of BCC in order to offer the reader not only the current diagnostic and therapeutic options, but also to show a glimpse of what the future has in store.
We believe that gene therapy is an important subject to be taken into consideration for future cancer studies and it's purpose at the end of the article is to stimulate the inquisitiveness of a researcher and to provide further research perspectives.
